# Regular perturbation on the group-velocity dispersion parameter for nonlinear fibre-optical communications

Vinícius Oliari [1], Erik Agrell[2 ✉] & Alex Alvarado [1]

Communication using the optical fibre channel can be challenging due to nonlinear effects that arise in the optical propagation. These effects represent physical processes that originate from light propagation in optical fibres. To obtain fundamental understandings of these processes, mathematical models are typically used. These models are based on approximations of the nonlinear Schrödinger equation, the differential equation that governs the propagation in an optical fibre. All available models in the literature are restricted to certain regimes of operation. Here, we present an approximate model for the nonlinear optical fibre channel in the weak-dispersion regime, in a noiseless scenario. The approximation is obtained by applying regular perturbation theory on the group-velocity dispersion parameter of the nonlinear Schrödinger equation. The proposed model is compared with three other models using the normalized square deviation metric and shown to be significantly more accurate for links with high nonlinearities and weak dispersion.

[1] Information and Communication Theory (ICT) Lab, Signal Processing Systems (SPS) Group, Department of Electrical Engineering, Eindhoven University of Technology, 5600 MB, Eindhoven, The Netherlands. [2] Department of Electrical Engineering, Chalmers University of Technology, Gothenburg SE-41296, Sweden. ✉email: agrell@chalmers.se

Optical fibre propagation can be modelled by the non-linear Schrödinger equation (NLSE)[1]. The NLSE is a partial differential equation that has three main effects: attenuation, second order dispersion, and Kerr nonlinearity. When considering the three effects together, the NLSE has no analytical solution for an arbitrary input pulse. The solution is obtained via a numerical method known as split step Fourier method (SSFM)[1,2]. The SSFM divides the fibre in small segments, known as steps, and increasing the number of steps increases the method's accuracy[2,3]. For a high number of steps, the method's computational complexity is a limiting factor[4]. To overcome this limitation, analytical models are generally used.

An analytical model that is easily mathematically manipulable is highly desirable as it can be used for improving fibre-optical transmission. As models are approximations of the NLSE, they can be used to build improved receivers and mitigate fibre effects[5–8], design signal shaping and coding[9,10], to estimate channel capacity[10–14], and even to predict system performance[15,16]. An extensive review on optical channel models can be found in ref. [10]. Each model will have a regime of operation based on the approximation used to derive it. The regimes classify models with respect to the group-velocity dispersion parameter $\beta_2$ (which in this paper we refer to as the linear coefficient), the Kerr nonlinear coefficient $\gamma$, and the input power.

Some of the main regimes of operation and models present in the literature are schematically represented in Fig. 1. If both the linear and nonlinear coefficients are zero, the fibre degenerates to an additive Gaussian noise (AWGN) channel (in the presence of noise), and no interesting effects of the fibre propagation appear. One of the simplest regimes that accounts for fibre propagation effects is when the linear coefficient is zero ($\beta_2 = 0$), represented by the green region ② of Fig. 1. In this case, the NLSE is modelled by a nonlinear phase shift[1], known as the nonlinear phase noise (NLPN) model[11,17]. The assumption of a zero linear coefficient results in a memoryless channel. This assumption was used in the literature, for example, when chromatic dispersion is completely compensated[18]. The same premise was considered when analysing the highly nonlinear regime, i.e., when the nonlinearities are predominant and may be an important effect to take into account[12,19]. As will be seen in the Results section, even in that regime the dispersion can deteriorate the performance of the model.

Another simple model arises when the nonlinear coefficient is zero ($\gamma = 0$), represented by the blue region ① of Fig. 1. In this case, the NLSE again admits an exact solution given by the so-called dispersion-only model[1]. This model considers the fibre propagation as an all-pass filter, whose phase response grows with the square of the frequency. As the model considers zero non-linearities, it is ideal for low power regimes, where the dispersion is the major effect.

For high power regimes, if the nonlinear coefficient is low but nonzero, the dispersion-only model becomes inaccurate, as will be shown in the Results section. In such scenario, regular per-turbation (RP) theory on the nonlinear coefficient[20–22] becomes a more suitable model, represented by the yellow region ③ in Fig. 1. The nonlinearities depend on the signal times the square of the absolute value of the signal, as will be seen in the nonlinear term of Eq. (3). This dependence makes the nonlinearities grow with the cube of the signal power, thus compromising the accuracy of the RP for high powers[20,21]. As will be shown in the Results section, RP on the nonlinear coefficient is accurate for a wider range of powers than the dispersion-only model. This wider range allows the RP model to model many communication systems[8,22]. The assumption of low nonlinearity is also used for other models, such as logarithmic perturbation[21] and Volterra series[23]. With respect to the Volterra series model, it was proved in ref. [20] that its $(2n+1)$-th order is equivalent to the $n$-th order RP on the nonlinear coefficient model.

The mentioned models only cover regions ①, ②, and ③ of Fig. 1. Models for the red region ④ and the brown region ⑤ do not exist in the published literature. The latter represents regimes where both the linear and nonlinear coefficient are high, and might not be achievable by perturbation models as, by the RP's definition, one of the coefficients should be low. In region ④, the absolute value of the linear coefficient is low; therefore, per-forming a RP is a reasonable approach.

In this paper, we propose a perturbation on the linear coeffi-cient of the NLSE, providing a model for the weak-dispersion regime represented by region ④. The proposed RP on the linear coefficient covers regimes where the nonlinear coefficient is high, and in contrast to the NLPN model, small amounts of dispersion are allowed. The RP on the linear coefficient is a model in closed mathematical form, and depends on the input field and the fibre parameters. A closed-form equation is derived for the continuous-time fibre output with single polarization. The pro-posed model is compared with the RP on the nonlinear coeffi-cient, the dispersion only, and the NLPN models. For comparison purposes, the fibre and simulation parameters (such as bandwidth and span length) are varied to identify regimes where each model is accurate. As will be shown in this paper, the RP on the linear coefficient is accurate for a wider range of powers than the RP on the nonlinear coefficient in low accumulated dispersion systems.

## Results

**Fibre propagation and metrics.** The noiseless propagation of the optical field $E$ at the retarded time frame $t$ and distance $z$ for a single polarization in a single-mode fibre can be represented by the NLSE[1]

$$\frac{\partial E(t,z)}{\partial z} = -\frac{\alpha}{2}E(t,z) - \frac{j\beta_2}{2}\frac{\partial^2 E(t,z)}{\partial t^2} + j\gamma|E(t,z)|^2 E(t,z), \quad (1)$$

where $\alpha$ is the attenuation coefficient, $\beta_2$ the group-velocity dis-persion parameter, and $\gamma$ the nonlinear coefficient. When nor-malizing the field $E$ via

$$E(t,z) = A(t,z)\mathrm{e}^{-\frac{\alpha}{2}z}, \quad (2)$$

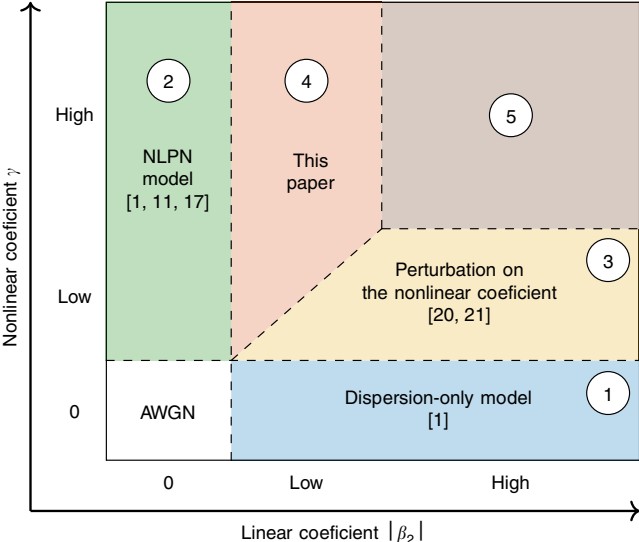

**Fig. 1 Validity region for different models for the nonlinear Schrödinger equation.** Each region is characterized by a combination of $|\beta_2|$ and $\gamma$ values. The models are derived by using approximations based on the magnitude of these two parameters.

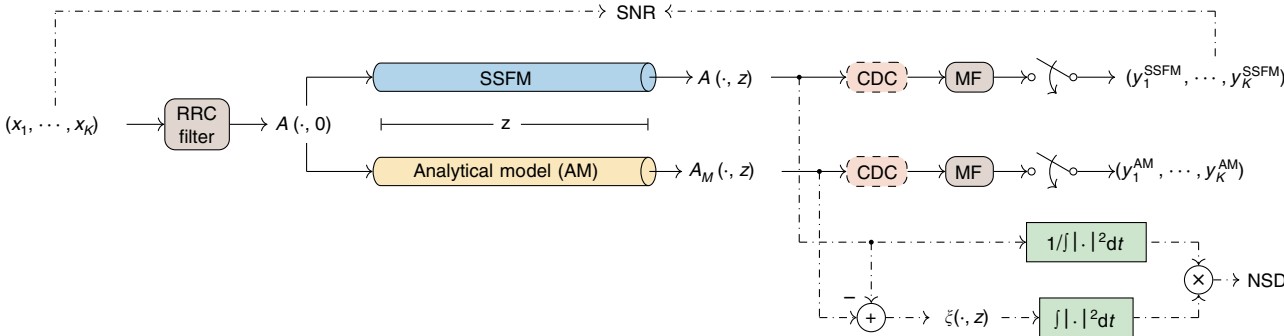

**Fig. 2 Schematic of NSD and SNR calculations.** The transmitted symbols $(x_1, \ldots, x_K)$ are filtered by a root-raised-cosine (RRC) filter, originating the input signal $A(\cdot, 0)$. This input signal is processed either by the SSFM or by an analytical model (AM), resulting in the outputs $A(\cdot, z)$ and $A_M(\cdot, z)$, respectively. The NSD is obtained from the SSFM output $A(\cdot, z)$ and a model output $A_M(\cdot, z)$, whereas the SNR is obtained from the transmitted symbols $(x_1, \ldots, x_K)$ and with the received symbols $(y_1, \ldots, y_K)$ from the SSFM or from an analytical model. To obtain the received symbols, the output waveforms are submitted to an optional chromatic dispersion compensation (CDC) block, followed by a matched filter (MF) and sampling.

Eq. (1) is simplified to

$$\frac{\partial A(t,z)}{\partial z} = \underbrace{-\frac{j\beta_2}{2}\frac{\partial^2 A(t,z)}{\partial t^2}}_{\text{linear term}} + \underbrace{j\gamma e^{-\alpha z}|A(t,z)|^2 A(t,z)}_{\text{nonlinear term}}. \quad (3)$$

By writing the NLSE as Eq. (3), the RP method can be more easily applied.

This version of the NLSE disregards dual-polarization effects, such as polarization-mode dispersion, as well as other effects like stimulated Raman scattering. The latter can be reasonably ignored given the low-bandwidth scenarios studied in this paper[1]. We chose to analyse the single-polarization equation as a first step toward RP on $\beta_2$. Nevertheless, single-polarization transmission is still attractive for low cost optical systems[24].

The first and second terms on the right-hand side of Eq. (3) are linear and nonlinear terms in the NLSE. In this paper, for simplicity, we will refer to them as linear and nonlinear terms, respectively, even though they refer to a normalized version of Eq. (1). When both the linear and nonlinear terms are considered together, there is no analytical solution for Eq. (1) for an arbitrary input pulse $A(\cdot, 0)$. However, by setting $\beta_2 = 0$ or $\gamma = 0$, it is possible to obtain simple analytical solutions[1]. These solutions are the basis of the models described below, and each of them has a regime where they can predict well the NLSE solution of Eq. (3).

In this paper, to quantify how well a model approximates the solution of the NLSE in Eq. (3), we will use a metric that relates the error between two waveforms: the normalized square deviation (NSD) (previously used in ref. [20]). The output of the SSFM algorithm will be considered as the solution $A$ of Eq. (3). The NSD calculation between $A(\cdot, z)$ and its approximation made by a certain model $A_M(\cdot, z)$ is illustrated in Fig. 2. For a certain propagation distance $z > 0$, the error $\xi$ between the model and the fibre output $A$ is

$$\xi(t,z) = A_M(t,z) - A(t,z). \quad (4)$$

Based on Eq. (4), we define the NSD as

$$\text{NSD} \triangleq \frac{\int_{-\infty}^{\infty} |\xi(t,z)|^2 \, dt}{\int_{-\infty}^{\infty} |A(t,z)|^2 \, dt}. \quad (5)$$

The NSD captures the average of the squared absolute error over the time dimension, which corresponds to the error energy. To enable a fair comparison between different input powers, the NSD is normalized by the energy of the fibre output $A(\cdot, z)$. Following ref. [21], we will use a threshold of 0.1% for comparing models. We will say that a model is precise if it gives an NSD below 0.1%.

In addition to the NSD, which characterizes the continuous-time performance, we also observe the discrete-time output, for which the signal-to-noise ratio (SNR) is a suitable metric. As shown in Fig. 2, we transmit a sequence of $K$ symbols $(x_1, \cdots, x_K)$, shaped by a root-raised-cosine (RRC) filter. The receiver consists of an optional chromatic dispersion compensation (CDC) block, followed by a matched filter and sampling. Even though no noise is added to the system, the received symbols are not exactly the transmitted ones. This difference is owing to limitations on the linear receiver we assume, which cannot undo the fibre propagation effects on the signal. Therefore, the SNR only accounts for signal–signal interactions in this case.

The SNR for a constellation with $M$ symbols is defined as

$$\text{SNR} \triangleq \frac{\sum_{m=1}^{M} |\bar{y}_m|^2}{\sum_{m=1}^{M} \frac{1}{N_m} \sum_{k=1}^{N_m} |y_{km} - \bar{y}_m|^2}, \quad (6)$$

where $\bar{y}_m = \frac{1}{N_m}\sum_{k=1}^{N_m} y_{km}$ is the average received symbol corresponding to the $m$-th constellation point, $N_m$ is the number of times the $m$-th constellation point was transmitted, and $y_{km}$ is the $k$-th received symbol given a fixed transmitted $m$-th constellation point. This SNR calculation assumes that we know the corresponding transmitted symbol for a given received symbol, and that the mean values $\bar{y}_m$ would be the signal components with ideal reception.

Notation convention: throughout this paper, we use $A(\cdot, z)$ to represent the evaluation of a two-variable function $A$ that depends on the retarded time frame, evaluated at distance $z$. In other words, we use this notation to emphasize that $A(\cdot, z)$ is still a function of the retarded time frame. The complex conjugate of $A$ is denoted by $A^*$. $\Re\{\cdot\}$ and $\Im\{\cdot\}$ give the real and imaginary parts of a complex number, respectively. Operators are denoted by calligraphic letters.

The numerical examples we will present investigate the limits and the operational regimes of each model. To vary the group-velocity dispersion, two types of fibre were considered: standard single-mode fibre (SSMF) and nonzero dispersion-shifted fibre (NZDSF). The SSMF has $\alpha = 0.2$ dB km$^{-1}$, $\beta_2 = -21.67$ ps$^2$ km$^{-1}$, and $\gamma = 1.2$ W$^{-1}$ km$^{-1}$, whereas the considered NZDSF has $\alpha = 0.22$ dB km$^{-1}$, $\beta_2 = -5.42$ ps$^2$ km$^{-1}$, and $\gamma = 1.46$ W$^{-1}$ km$^{-1}$. Except in the symbol rate variation analysis, all the simulations consider a symbol rate of 10 Gbaud. The modulation format is 64-ary quadrature amplitude modulation (64-QAM) unless otherwise stated.

In what follows, we review 3 models available in the literature and then, present the RP on $\beta_2$.

**Dispersion-only model**. When considering $\gamma = 0$, Eq. (3) simplifies to[1]

$$\frac{\partial A_M(t,z)}{\partial z} = -\frac{j\beta_2}{2}\frac{\partial^2 A_M(t,z)}{\partial t^2}, \qquad (7)$$

which has the exact solution[25]

$$A_M(t,z) = (A(\cdot,0) * h(\cdot,z))(t) = \mathcal{D}_z\{A(\cdot,0)\}(t). \qquad (8)$$

In Eq. (8), $*$ represents convolution, $h$ is given by

$$h(t,z) = \frac{1}{\sqrt{j2\pi\beta_2 z}}e^{-\frac{j}{2\beta_2 z}t^2}, \qquad (9)$$

and $\mathcal{D}_z$ is the dispersion operator defined as

$$\mathcal{D}_z\{f\}(t) \triangleq (f * h(\cdot,z))(t), \qquad (10)$$

where $f$ is a function of $t$. The solution in Eq. (8) is called the dispersion-only model, and is a linear, time-invariant all-pass filter. It corresponds to ① in Fig. 1.

*Example* 1. Figure 3a shows the NSD vs. input power for the dispersion-only model for the SSMF and an NZDSF. As shown in Fig. 3a, the model is an accurate approximation of the fibre output in this system for powers lower than $+2$ dBm for the 10 km fibre. This is as a regime where the nonlinearities are not predominant. As the input power increases, the NSD increases by $\sim 2$ dB/dBm. This figure also shows that a change in $\beta_2$ of approximately four times does not considerably change the NSD. By increasing the distance from 10 to 80 km, the NSD grows almost one order of magnitude, which can be justified by the extended interaction between the nonlinearities and the chromatic dispersion, not modelled in this solution.

**NLPN model**. For $\beta_2 = 0$, Eq. (3) simplifies to[1]

$$\frac{\partial A_M(t,z)}{\partial z} = j\gamma e^{-\alpha z}|A_M(t,z)|^2 A_M(t,z), \qquad (11)$$

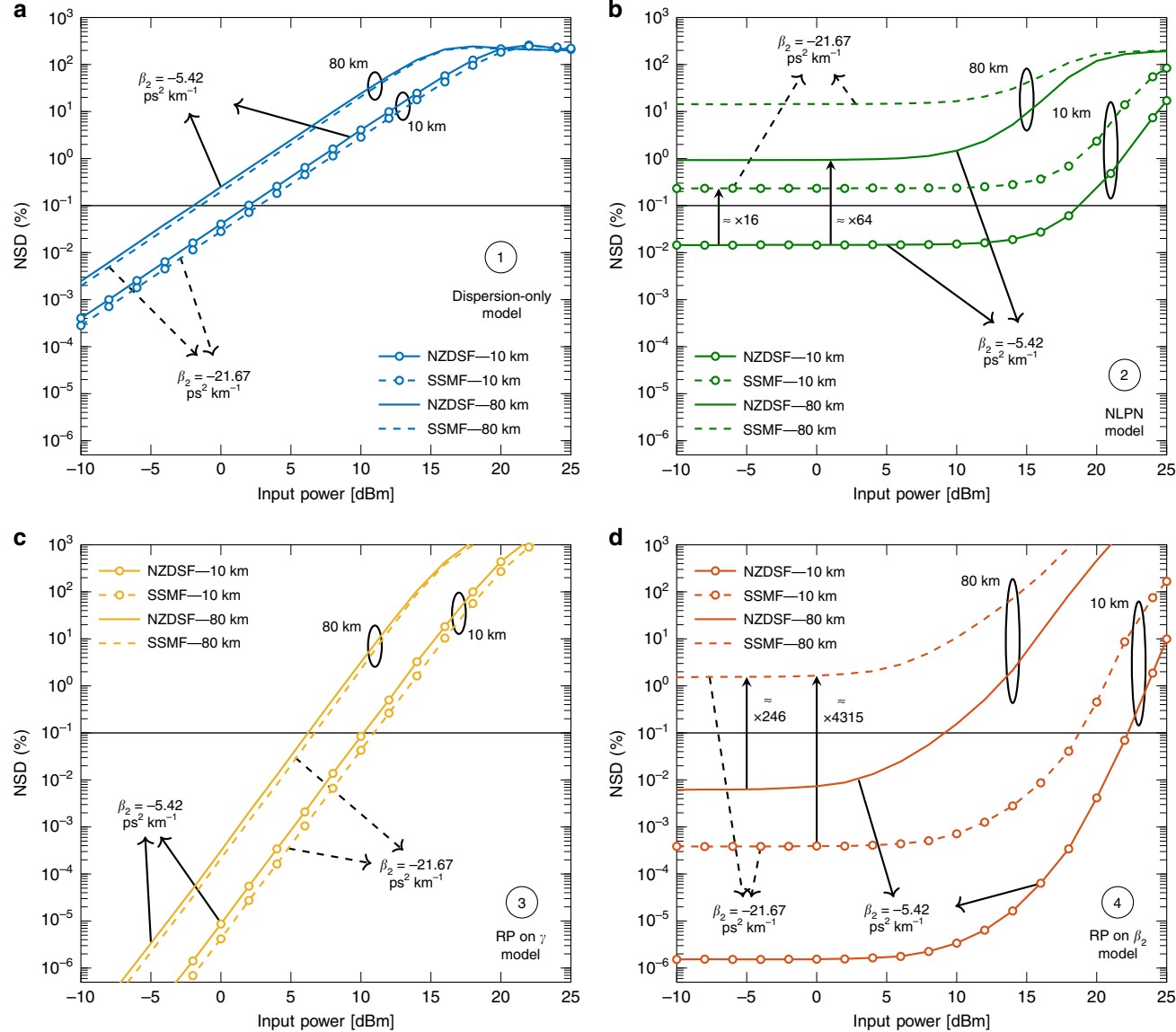

**Fig. 3 NSD for the four described models as a function of the input power.** Two fibre lengths (10 and 80 km) and two types of fibre (NZDSF and SSMF). **a** Dispersion-only model given by (8). **b** NLPN model given by (12). **c** RP on $\gamma$ model given by (15). **d** RP on $\beta_2$ model given by (19). Each model has a different accuracy, measured in NSD, for a specific set of input powers, $\beta_2$ coefficient, and distance. The black horizontal line represents a constant 0.1% NSD, which is used for comparison between models.

which has the exact solution

$$A_M(t,z) = A(t,0)e^{j\gamma|A(t,0)|^2 G(z)}, \qquad (12)$$

where

$$G(z) = \int_0^z e^{-\alpha u}\, du = \frac{1 - e^{-\alpha z}}{\alpha} \qquad (13)$$

is the effective length. The solution in Eq. (12) is called the NLPN model, and is a memoryless, signal-dependent phase shift. This model is ② in Fig. 1.

*Example* 2. Figure 3b shows the NSD of the NLPN model for the same scenario as Example 1. For the NLPN, increasing the value of $|\beta_2|$ (changing from NZDSF to SSMF) can deteriorate the NSD by a factor of almost 10 times for the same distance at 0 dBm. This effect can be justified by the increased dispersion contribution to the signal, not modelled by this solution. Furthermore, the accumulated dispersion is also increased when extending the fibre length from 10 to 80 km, which increases the NSD by two orders of magnitude. This last increase can also be seen by the fact that, at long distances, $G(z) \approx 1/\alpha$ and Eq. (12) barely changes with respect to $z$. However, this result is not consistent with the actual fibre output, which still depends on the distance $z$ in the presence of chromatic dispersion. Although only the 10 km NZDSF presented an NSD below the threshold of 0.1%, the NLPN model has an NSD almost constant for powers lower than 10 dBm.

**RP on $\gamma$.** The RP method consists of representing the solution of an equation with an expansion in terms of simplified solutions and a coefficient. In the RP on $\gamma$, the general solution can be written as[20,22]

$$A(t,z) = \sum_{k=0}^{\infty} \gamma^k A_k(t,z), \qquad (14)$$

where the functions $A_k$ are functions that depend on the initial field $A(\cdot, 0)$. In the first order RP on $\gamma$, Eq. (14) is truncated at $k = 1$, and the functions $A_0$ and $A_1$ are given in the next theorem. The result of Theorem 1 is well known in the literature (see ref. [20–22]). However, we include its proof in Supplementary Note 2 for consistency with the notation.

*Theorem* 1. Let $A$ be the solution of the NLSE in Eq. (3) with initial condition $A(\cdot, 0)$. Then, $A$ can be approximated by $A_M$, the first order RP on the nonlinear coefficient $\gamma$ of Eq. (3), written as

$$A_M(t,z) = A_0(t,z) + \gamma A_1(t,z), \qquad (15)$$

where

$$A_0(t,z) = \mathcal{D}_z\{A(\cdot,0)\}(t) \qquad (16)$$

is the dispersion-only solution in Eq. (8) and

$$A_1(t,z) = j\int_0^z e^{-\alpha u} \mathcal{D}_{z-u}\{|A_0(\cdot,u)|^2 A_0(\cdot,u)\}(t)\, du. \qquad (17)$$

The proof is given in Supplementary Note 2.

*Example* 3. In Fig. 3c, the NSD results for SSMF and NZDSF fibres are presented. This figure shows that the NSD scales as 4 dB/dBm, which is twice as much as the value found for the dispersion-only model example (see Fig. 3a). This behaviour might be explained by the cubic signal power dependence of the nonlinear term and $A_1$ (see nonlinear term in Eq. (3)). Despite this faster growth of the NSD, the crossing point of the curves with the 0.1% threshold happens at higher powers compared with the dispersion-only case. This shows that the RP on $\gamma$ model has a wider range of validity than the dispersion-only model. Figure 3c also shows that, for the RP on $\gamma$ model, just like for the dispersion-only case, increasing $|\beta_2|$ slightly improves the NSD.

The increase in the fibre length also deteriorates the performance of the RP on $\gamma$. As expected, the NSD grows when the power of the input signal is increased. To reduce the approximation error, more terms of the expansion in Eq. (14) can be considered. This comes at the cost of a higher model complexity, as done in ref. [20].

**Proposed model: RP on $\beta_2$.** We now present the proposed model, which was derived based on the RP method to accurately represent the NLSE in the highly nonlinear regime, illustrated by region ④ in Fig. 1.

In analogy with the RP on $\gamma$, the RP method can be applied to $\beta_2$. The only difference is that now the expansion of $A$ is written in terms of $\beta_2$ as

$$A(t,z) = \sum_{k=0}^{\infty} \beta_2^k A_k(t,z). \qquad (18)$$

The terms $A_0$ and $A_1$ for the first order RP on $\beta_2$ are given in the next theorem, which is the main contribution of this paper.

*Theorem* 2. Let $A$ be the solution of the NLSE in Eq. (3) with initial condition $A(\cdot, 0)$. Then, $A$ can be approximated by $A_M$, the first order RP on the linear coefficient $\beta_2$ of Eq. (3), written as

$$A_M(t,z) = A_0(t,z) + \beta_2 A_1(t,z), \qquad (19)$$

where

$$A_0(t,z) = A(t,0)e^{j\gamma|A(t,0)|^2 G(z)}, \qquad (20)$$

and

$$A_1(t,z) = B(t,z)e^{j\gamma|A(t,0)|^2 G(z)}, \qquad (21)$$

with $B$ given by

$$\begin{aligned} B(t,z) = &-M(t)z + G_1(z)R(t) + G_2(z)P(t) \\ &- 2j\gamma A(t,0)\Re\{A^*(t,0)V(t,z)\}, \end{aligned} \qquad (22)$$

$$\begin{aligned} V(t,z) = &\,G(z)[M(t)z - G_1(z)R(t) - G_2(z)P(t)] \\ &- G_1(z)M(t) + G_2(z)R(t) + G_3(z)P(t), \end{aligned} \qquad (23)$$

$$M(t) = \frac{j}{2}\frac{\partial^2 A(t,0)}{\partial t^2}, \qquad (24)$$

$$R(t) = \frac{\gamma}{2}A(t,0)\frac{\partial^2|A(t,0)|^2}{\partial t^2} + \gamma\frac{\partial A(t,0)}{\partial t}\frac{\partial|A(t,0)|^2}{\partial t}, \qquad (25)$$

$$P(t) = \frac{j\gamma^2}{2}A(t,0)\left(\frac{\partial|A(t,0)|^2}{\partial t}\right)^2, \qquad (26)$$

$$G_1(z) = \frac{\alpha z + e^{-\alpha z} - 1}{\alpha^2}, \qquad (27)$$

$$G_2(z) = \frac{2\alpha z + 4e^{-\alpha z} - e^{-2\alpha z} - 3}{2\alpha^3}, \qquad (28)$$

$$G_3(z) = \frac{6\alpha z + 18e^{-\alpha z} - 9e^{-2\alpha z} + 2e^{-3\alpha z} - 11}{6\alpha^4}. \qquad (29)$$

The proof is postponed to Supplementary Note 3.

By comparing Eq. (19) and Eq. (20) with Eq. (12), it is clear that the RP on $\beta_2$ corresponds to the NLPN solution perturbed by the dispersion. The perturbation term $A_1$ in Eq. (21) is in closed form and it depends on the derivatives of the input field $A(\cdot, 0)$. Note also that both $A_1$ and $A_0$ are multiplied by the same phase rotation, as shown in Eq. (20) and Eq. (21).

The functions $A_0$ and $A_1$ depend on elementary functions of $z$ (see equations (13), (27), (28), and (29)). This results in the same calculation time for $A_M$ at any distance $z$, in contrast to the SSFM, for which the number of necessary steps increases with the distance.

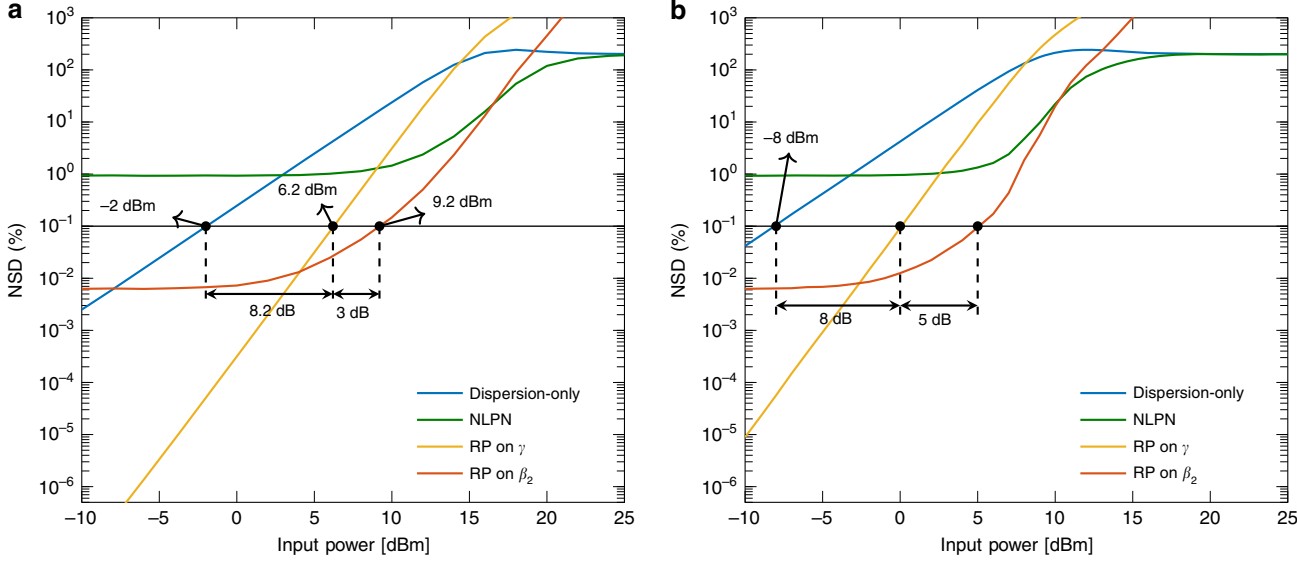

**Fig. 4 NSD for the four models in 80 km of an NZDSF. a** With attenuation ($\alpha = 0.22$ dB km$^{-1}$). **b** Without attenuation ($\alpha = 0$ dB km$^{-1}$). The power gap in dB between the NSD curves is related to different input powers that allow each model to achieve 0.1% NSD. In the system without attenuation **b**, this gap is increased when comparing RP on $\gamma$ with RP on $\beta_2$.

For this reason, RP on $\beta_2$ calculations are significantly faster than SSFM calculations for the parameters simulated in this paper.

*Example* 4. Figure 3d presents the NSD for the RP on $\beta_2$ for the SSMF and the NZDSF. As shown in Fig. 3d, the NSD is approximately constant below a certain input power (0 dBm in this case). In addition, changes in the amount of dispersion severely impact the NSD. For example, at 0 dBm, increasing $\beta_2$ (going from NZDSF to SSMF) in the 80 km system, the NSD rises more than two orders of magnitude. This model is also very sensitive to the distance. Increasing the fibre length from 10 to 80 km in the SSMF system, the NSD rises more than three orders of magnitude for 0 dBm.

As discussed above, the RP on $\beta_2$ in Eq. (19) is the sum of the NLPN solution ($A_0$) and a term accounting for the dispersion ($\beta_2 A_1$). Therefore, some similarities in the NSD curves of these two models are expected (in analogy to the RP on $\gamma$ and the dispersion-only model, discussed in Example 3). By comparing Fig. 3d with Fig. 3b, the RP on $\beta_2$ is an improved version of the NLPN model, just like the RP on $\gamma$ is an improved version of the dispersion-only model.

*Example* 5. Figure 4a presents a comparison of the four models for an 80 km NZDSF. As shown in Fig. 4a, only the dispersion-only, the RP on $\gamma$, and the RP on $\beta_2$ models have NSDs below the threshold of 0.1% for some powers. The dispersion-only model crosses the threshold at $-2$ dBm, whereas the RP on $\gamma$ crosses it at 6.2 dBm, presenting a gain of 8.2 dB. The RP on $\beta_2$ crosses the threshold at even higher powers (9.2 dBm), with a gain of 3 dB with respect to the RP on $\gamma$.

In systems with distributed Raman amplification, the power profile is approximately flat[26] and the attenuation term in (1) is often neglected. In this case, a simpler analytical form for $B(t, z)$ when compared with Eq. (22) is achieved. This simplification is given in the next theorem.

*Theorem* 3. With ideal distributed amplification, the functions $A_0$ and $A_1$ in Eq. (20) and Eq. (21) can be written as

$$A_0(t,z) = A(t,0)e^{j\gamma|A(t,0)|^2 z}, \tag{30}$$

$$A_1(t,z) = B(t,z)e^{j\gamma|A(t,0)|^2 z}, \tag{31}$$

where

$$
\begin{aligned}
B(t,z) = & -M(t)z + \frac{z^2}{2}R(t) + \frac{z^3}{3}P(t) \\
& - 2j\gamma A(t,0)\Re\left\{A^*(t,0)\left[\frac{z^2}{2}M(t) - \frac{z^3}{6}R(t) - \frac{z^4}{12}P(t)\right]\right\}
\end{aligned}
\tag{32}
$$

and where $M(t)$, $R(t)$, and $P(t)$ are given by Eq. (24), Eq. (25), and Eq. (26), respectively. The proof is postponed to Supplementary Note 4.

*Example* 6. Figure 4b shows the NSD for the four models with the same parameters as in Example 5, except that in this case there is no attenuation. For powers below $-5$ dBm, the NLPN model and the RP on $\beta_2$ present almost the same NSD compared with Fig. 4a. This behaviour could be justified by the small impact of the nonlinearities for low powers. The NSD values in that regime become close to each other, since the attenuation is mostly connected to the nonlinear effect (as can be seen in Eq. (3)). The curves cross the threshold at lower powers compared with Fig. 4a, excluding the NLPN model, which remains above the threshold for all analysed powers. We believe that the lower threshold crossing happens owing to the additional interactions between nonlinearities and dispersion present in the lack of attenuation. Although the RP on $\gamma$ crosses the threshold at 0 dBm, the RP on $\beta_2$ crosses at 5 dBm, representing a gain of 5 dB (2 dB more compared with the attenuation case).

In the previous examples, the parameters $\gamma$ and $\beta_2$ were fixed, along with the simulation bandwidth and the fibre length. These parameters are further investigated in the next examples, where the four models are compared with each other in systems with attenuation. For the next simulations, most of the parameters given in the previous examples are still considered; however, some of them will be changed. We will discuss fibre length variation, symbol rate variation, and $\beta_2$ and $\gamma$ variation.

**Variation of $\beta_2$ and $\gamma$.** To analyse the impact of changes in $\beta_2$ and $\gamma$, we consider a fibre length of 80 km with fixed input power of 5 dBm, symbol rate of 10 Gbaud, and $\alpha = 0.2$ dB km$^{-1}$. In this analysis, only the RP on $\gamma$ and $\beta_2$ will be considered. Figure 5a

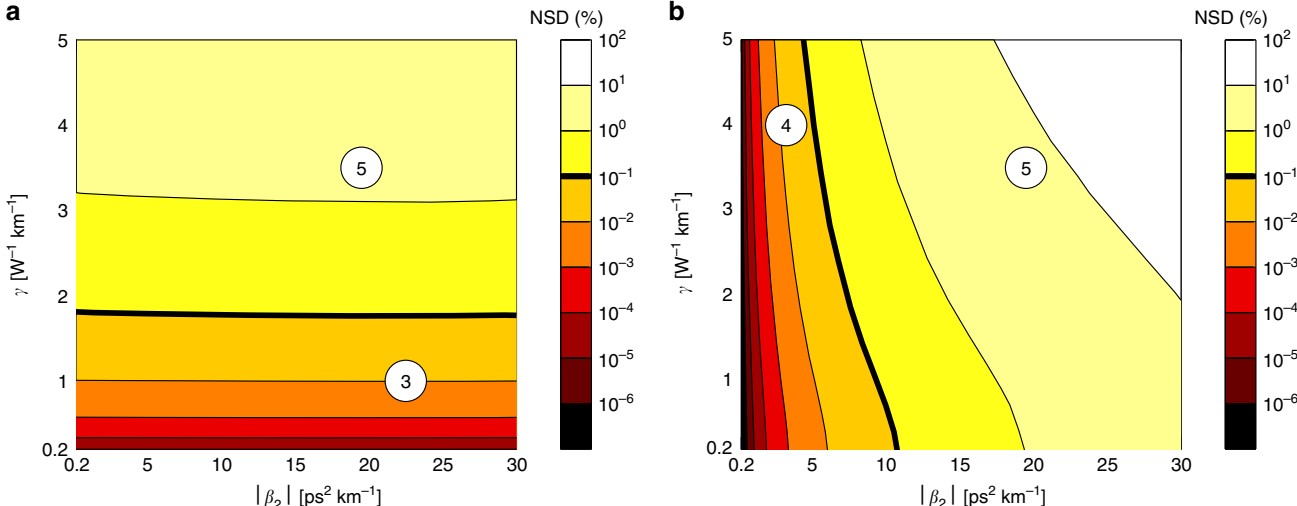

**Fig. 5 NSD for the RP on $\gamma$ and the RP on $\beta_2$ for different values of $|\beta_2|$ and $\gamma$.** The system consists of an 80 km fibre with fixed input power 5 dBm, symbol rate of 10 Gbaud, $\alpha = 0.2$ dB km$^{-1}$, and negative $\beta_2$. **a** Regular Perturbation on $\gamma$. **b** Regular Perturbation on $\beta_2$. Region ③ represents low $\gamma$ and large $|\beta_2|$ values (see Fig. 1). Region ④ represents low $|\beta_2|$ and large $\gamma$ values. Region ⑤ represents large $\gamma$ and large $|\beta_2|$ values. The thick black line of constant 0.1% NSD is used as a threshold for accuracy comparison. The results indicate, as expected, that RP on $\gamma$ is accurate for region ③ and RP on $\beta_2$ is accurate for region ④. Both models yield low accuracy in region ⑤.

shows the NSD for the RP on $\gamma$ for different values of $|\beta_2|$ (negative $\beta_2$) and $\gamma$. As depicted in Fig. 5a, variations in $\beta_2$ practically do not affect the accuracy of the model, whereas changes in $\gamma$ have a major impact. In analogy with Fig. 1, the lower region of Fig. 5a is equivalent to region ③, where the model is accurate (NSD < 0.1%). This region covers values of $\gamma$ of up to ~ 1.78 W$^{-1}$ km$^{-1}$ for this system. Making the same analysis for RP on $\beta_2$ leads to Fig. 5b, which shows the NSD for the same range of $\gamma$ and $\beta_2$ values. In this case, the area where the model is accurate is vertical, in analogy to region ④ in Fig. 1. The NSD for the RP on $\beta_2$ changes mostly with the value of $\beta_2$; however, changes in $\gamma$ can also significantly affect the performance, specially for high $|\beta_2|$ values. The intersection of the areas that are not accurate for any of the models is related to region ⑤ in Fig. 1.

**Fibre length variation**. For the fibre length variation analysis, we consider an NZDSF with fixed input power of 5 dBm and a symbol rate of 10 Gbaud. Figure 6a shows the NSD versus fibre length for the four different models. All the four models increase the NSD when increasing the fibre length; however, the dispersion-only and the RP on $\gamma$ seem to converge to a constant NSD value. This convergence is owing to the attenuation on the nonlinear term in Eq. (3): for high distances, the nonlinearities do not considerably affect the signal, and the major effect is the dispersion. As these two models fully predict the dispersion effect, they do not lose accuracy in that regime. The RP on $\beta_2$, for this system, can reach ~ 120 km within the NSD threshold of 0.1%, and for fibre lengths lower than 90 km, the model outperforms the RP on $\gamma$.

**Symbol rate variation**. For the symbol rate variation analysis, we consider 80 km of an NZDSF with fixed input power of 5 dBm. Figure 6b depicts the NSD variation with respect to the symbol rate. As shown in Fig. 6b, the dispersion-only model and the RP on $\gamma$ do not change their accuracy when varying the symbol rate. On the other hand, the NLPN model and the RP on $\beta_2$ drastically drop the NSD when decreasing the symbol rate. For bandwidths lower than 4 Gbaud, even the NLPN can outperform the RP on $\gamma$. This behaviour may be justified by the decreasing influence of the dispersion on the signal, since according to Eq. (7), higher

frequencies are more affected by dispersion owing to their high second derivative.

**Fibre length versus symbol rate**. The previous two sections analysed the NSD by varying the fibre length and symbol rate separately. Both of these parameters influence the accumulated dispersion. Therefore, given a fibre length or a symbol rate, we can find the values for the other parameter in which RP on $\beta_2$ is accurate[27].

Figure 7 depicts the NSD given a fibre length and a symbol rate for an NZDSF with fixed input power 5 dBm. As shown in Fig. 7, the model can accurately handle arbitrarily large fibre lengths if the symbol rate is small enough and vice versa. By fixing a fibre length of 80 km, the maximum symbol rate in which RP on $\beta_2$ is still accurate is 12.55 Gbaud (see triangular marker in Fig. 7). If the distance is reduced to 20 km, the symbol rate can be increased until 27.38 Gbaud (see square marker in Fig. 7). These values show that by reducing the fibre length by a factor of 4, the symbol rate can be increased by a factor of ~ 2.18. This difference in scaling factors was already expected considering that the accumulated chromatic dispersion increases linearly with the distance and with the square of the signal bandwidth.

The thick solid line in Fig. 7 at an NSD of 0.1% can be seen as a conservative threshold. Choosing other metrics, such as SNR, might motivate different conclusions. For example, we will show in the discrete-time performance section (ahead), RP on $\beta_2$ can have an SNR close to that of the SSFM, even though RP on $\beta_2$ yields an NSD in the order of 30% in that scenario.

**Modulation format impact**. The previous simulations were based on 64-QAM. For different modulation formats, the transmitted signal statistics change. As signal statistics impact the nonlinear effect, the NSD curves can be different for other modulation formats[27].

Figure 8 illustrates the NSD versus input power for three modulation formats on 80 km of an NZDSF with a symbol rate of 10 Gbaud. As shown in Fig. 8, quadrature phase-shift keying (QPSK) yields lower NSD than the other two modulation formats. We believe this behaviour is justified by the QPSK's high tolerance to nonlinearities, which reduces the error in the

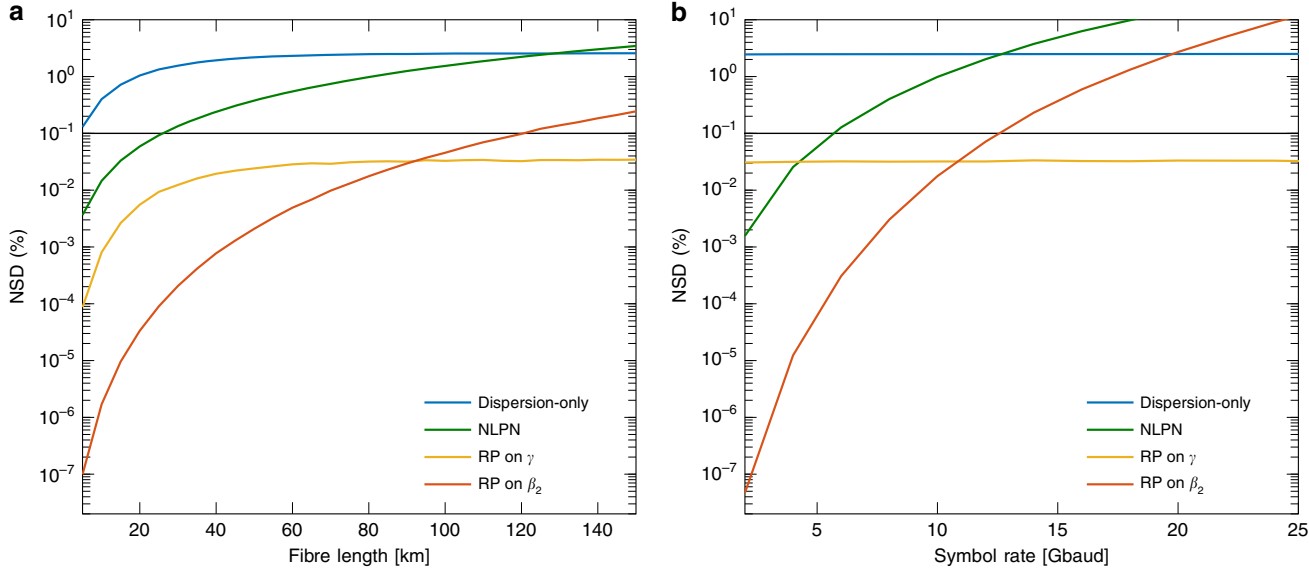

**Fig. 6 NSD versus fibre length and symbol rate for the four models.** The system consists of an NZDSF with fixed input power 5 dBm. **a** NSD versus fibre length at a symbol rate of 10 Gbaud. **b** NSD versus symbol rate for a fibre length of 80 km. Changes in fibre length or symbol rate result in a wider range of NSD variation for RP on $\beta_2$ than for RP on $\gamma$.

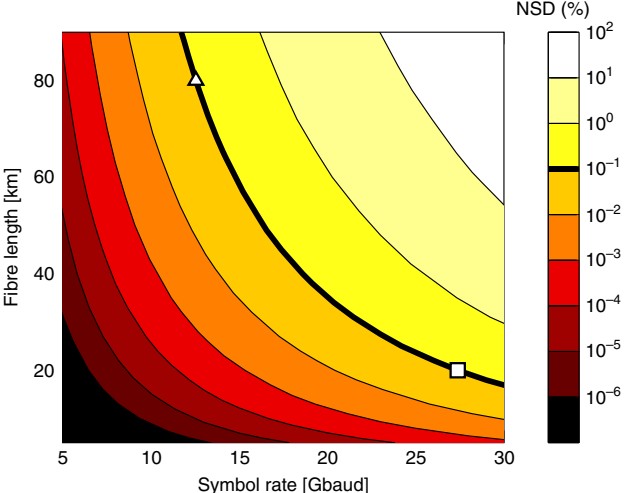

**Fig. 7 NSD for RP on $\beta_2$ for different values of fibre length and symbol rate.** The system consists of a 80 km NZDSF with fixed input power 5 dBm. The triangular marker on the top of the figure is associated with the distance of 80 km and symbol rate of 12.55 Gbaud. The square marker on the bottom right is associated with the distance of 20 km and symbol rate of 27.38 Gbaud. Both markers lie on the thick black line representing a constant 0.1% NSD.

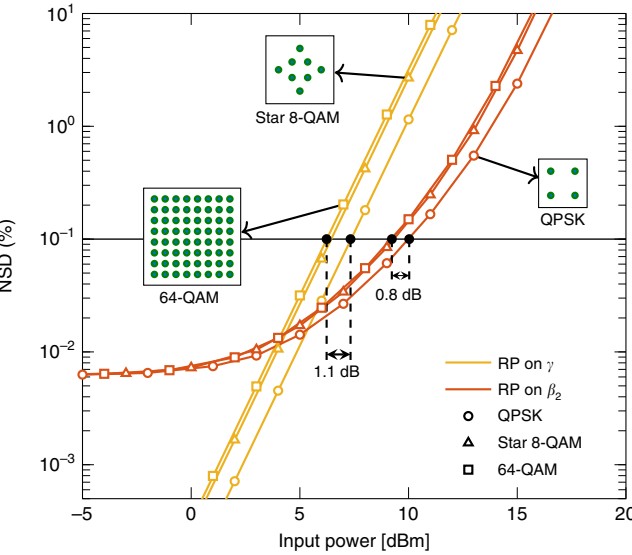

**Fig. 8 Impact of QPSK, star 8-QAM, and 64-QAM modulation formats on the NSD versus input power.** The system consists of an 80 km NZDSF with 10 Gbaud symbol rate. The modulation format impact is measured by the power gap in dB between the crossing points of the NSD curves for each modulation format with the horizontal black curve representing a constant 0.1% NSD.

first-order RP approximation. The performance of star 8-QAM and 64-QAM are almost the same for the considered input powers. The gap between QPSK and 64-QAM for the 0.1% threshold crossing is 0.8 dB for RP on $\beta_2$ and 1.1 dB for RP on $\gamma$. The higher gap for RP on $\gamma$ might indicate that this model is more sensitive to changes in the modulation format than RP on $\beta_2$.

**Discrete-time performance**. As discussed in the "Fibre propagation and metrics" section, the analysis of the received symbols might bring insightful conclusions about the models. In order to clearly visualize the fibre effects on the received constellations, the SNR simulations were based on QPSK modulation over a 20 km NZDSF. At the receiver, we considered two cases: with and

without CDC, followed by matched filtering and sampling (see Fig. 2). We emphasize that these operations at the receiver are applied to the continuous-time output of the models and SSFM, and we are not using discrete-time analytical models.

Figure 9a shows the SNR for SSFM, RP on $\gamma$, and RP on $\beta_2$ at 10 Gbaud for both receivers. As depicted in Figure 9a, for input powers lower than 6 dBm, the SNR for the receiver with CDC is higher than the one without CDC. The latter converges to ~35.9 dB. This convergence could be explained by the uncompensated dispersion effect, which does not depend on the signal power (a linear effect). For input powers >8 dBm, the systems with and without CDC are equivalent in SNR performance. This

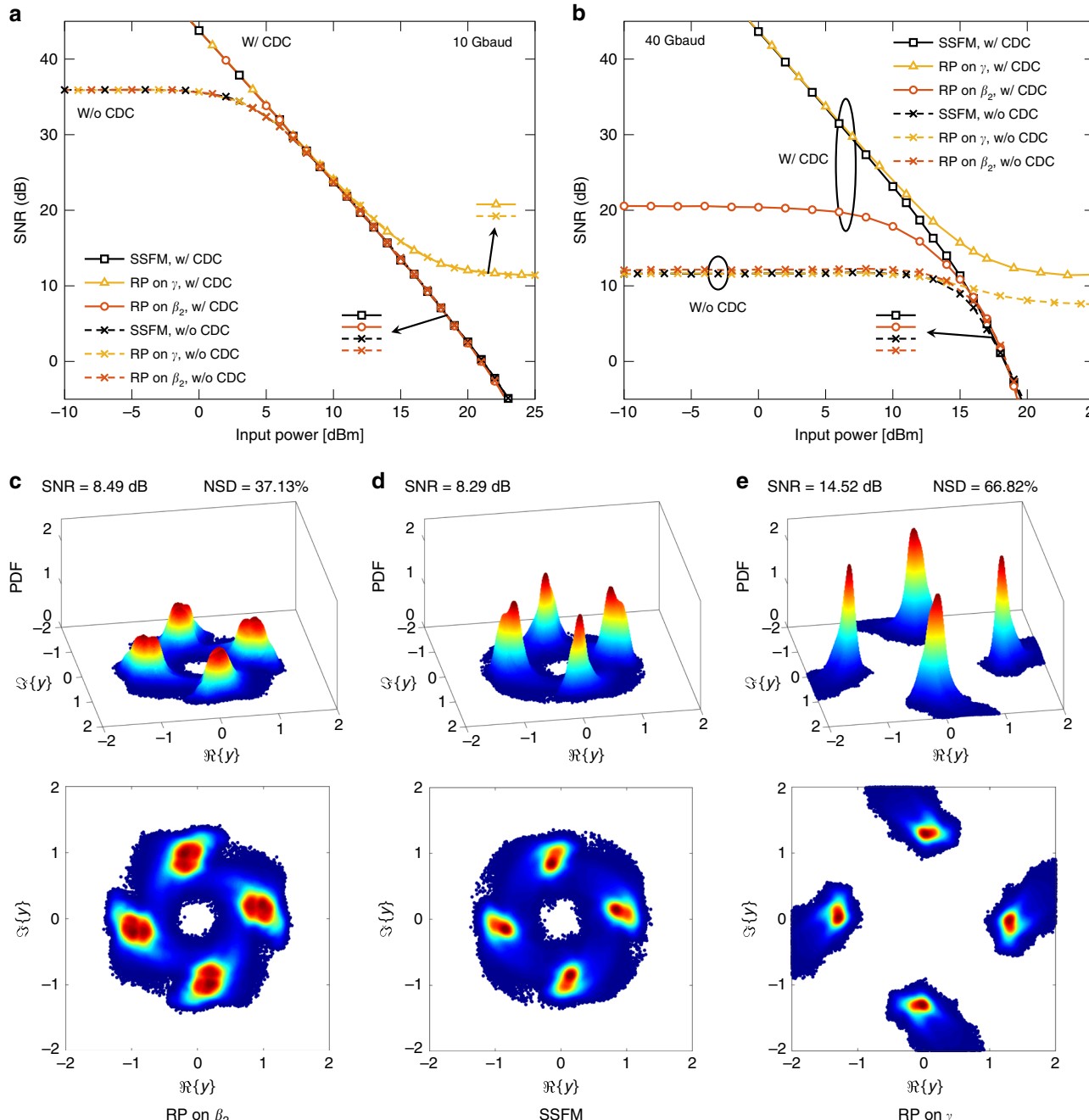

**Fig. 9 SNR versus input power, received constellations, and their respective PDFs.** On the SNR results, receivers with (w/) and without (w/o) CDC were considered for a QPSK modulation over a 20 km NZDSF. The constellations were obtained for 40 Gbaud with CDC at 16 dBm. **a** SNR results for the three models at a symbol rate of 10 Gbaud. **b** SNR results for the three models at 40 Gbaud. **c** Received RP on $\beta_2$ constellation and respective PDF. **d** Received SSFM constellation and respective PDF. **e** Received RP on $\gamma$ constellation and respective PDF. The arrows pointing to the markers in **a** and **b** represent which curves are overlapped in those parts of the figure. In **c**, **d**, and **e**, the respective SNR values are shown on top of each of them. For **c** and **e**, the NSD values are also displayed.

behaviour might be explained by the predominance of the nonlinear effect at these input powers. For input powers >11 dBm, the SNR is higher for RP on $\gamma$ than for SSFM and RP on $\beta_2$. In this power regime, RP on $\beta_2$ can be seen as a more accurate model for SNR calculations.

Figure 9b illustrates the SNR for SSFM, RP on $\gamma$, and RP on $\beta_2$ at 40 Gbaud for both receivers. At this symbol rate and input powers lower than 15 dBm, RP on $\beta_2$ with CDC does not follow the SNR of SSFM and RP on $\gamma$. This difference might be related to the precarious estimation of the dispersion effect by the RP on $\beta_2$

for this higher bandwidth scenario (see Fig. 7). With a poor estimation of the chromatic dispersion, the CDC will not compensate the exact dispersion effect predicted by the RP on $\beta_2$ model. For the system without CDC, the performance of RP on $\beta_2$ is close to the SSFM performance for all displayed input powers. The SNR for RP on $\gamma$ for systems with and without CDC diverges again from the SSFM performance at input powers higher than 11 and 16 dBm, respectively. Meanwhile, at input powers higher than 16 dBm, the performance of RP on $\beta_2$ for the CDC system approaches the SSFM performance.

For an input power of 16 dBm in the 40 Gbaud system with CDC, the received constellations for RP on $\beta_2$, SSFM, and RP on $\gamma$ are shown in Figs. 9c, d, and e, respectively, with their corresponding probability density functions (PDFs). As illustrated in these figures, the constellation for RP on $\beta_2$ approximately preserves the circular shape of the SSFM's constellation, which is not observed in the constellation for RP on $\gamma$. This preservation could be explained by the better prediction of nonlinear effects at this high input power by the RP on $\beta_2$. The received symbols for RP on $\gamma$ are mostly outside the unitary square given by $|y| \leq 1$, which shows a high gain of energy when using this model. In addition, the PDF for RP on $\gamma$ shows that its symbols are highly concentrated toward a single point, which was expected by its high SNR. Even though the SNR for RP on $\beta_2$ is close to the SNR of SSFM (8.49 and 8.29 dB, respectively), the received constellation shapes are slightly different. For example, as observed in the constellation PDFs, the RP on $\beta_2$ received symbols are more spread than the SSFM symbols. This contrast means that the SNR alone might not indicate precisely the accuracy of the model. On the other hand, a high NSD for RP on $\beta_2$ (37.13%) may not indicate that the received signal in discrete time is severely different from the reference given by the SSFM, showing that our proposed model is accurate in scenarios where nonlinearities are the dominant effect.

## Discussion

This paper presented a new closed form analytical approximation for the solution of the NLSE: the RP on $\beta_2$. The derived approximation is a suitable model for low symbol rates, low fibre lengths, and/or high input powers. The RP on $\beta_2$ was compared with three other models with respect to variations in the bandwidth, fibre length, input power, and fibre parameters.

The main comparison was with the RP on $\gamma$, a well-known model in the literature that is accurate in the regime of high dispersion and low nonlinearities. In a NZDSF of 80 km with attenuation, the RP on $\beta_2$ can be used as an accurate model until input powers of 9 dBm, whereas the RP on $\gamma$ is accurate only to input powers lower than 6 dBm. In addition, the RP on $\beta_2$ is accurate for high $\gamma$ values where the RP on $\gamma$ is not. Thus, the new proposed model is convenient for the opposite regime, where the nonlinearities are predominant, and the dispersion has a minor effect.

As all simplified models, the proposed model has a limited range of validity. At the moment, the main applicability of the model is for applications that rely on low bandwidths (below 11 GHz) and short distances (below 80 km). This includes, for example, passive optical networks[28]. Another short-distance low-bandwidth application is hybrid fibre coax systems[29]. The model is in its present form not intended for long-haul or wavelength-division-multiplexed systems.

Another drawback of the proposed model is that it neglects the noise. This effect has been considered in the literature for the RP on $\gamma$ in ref. [22], where noise was added in the zeroth order linear equation, followed by a Karhunen–Loève expansion to account for nonlinear signal–noise interactions. The same approach for the RP on $\beta_2$ would lead to cumbersome equations, as the zeroth order equation is nonlinear in this case.

This paper represents the first steps in the theory of the proposed model. Possible extensions of this work are designing a receiver based on the RP on $\beta_2$, deriving higher-order perturbations of the model, and adding noise within the RP analysis. The derivations were conducted for a single-polarization system, and the equations for dual-polarization are still a subject of further investigation. Separating the contribution of individual pulses and finding a discrete model are also left as future work. Although the focus of this paper was on optical-fibre communications, we believe that our model can be applied to other fields where the NLSE is applicable.

## Methods

**Simulation specifications**. The simulations were conducted in Matlab® and considered $2^{15}$ symbols randomly chosen from different constellations during the paper. The constellations were generated with unitary energy. To generate the constellation figures and PDFs in Fig. 9c, d, and e, we used $2^{20}$ symbols for a smoother plot. The colour for each received symbol was attributed from a color-map according to the PDF values. The symbols were oversampled by 16 samples per symbol. After oversampling, the signal was shaped by an ideal RRC filter, with roll-off factor of 0.1, implemented in the frequency domain. After filtering, the signal was scaled to adjust to the desired input power. The resulting waveform was used as the input of either the SSFM or one of the models. For the SSFM case, we considered a symmetric SSFM implementation, with step-size 0.1 km. The step-size and the simulation bandwidth substantially impact the SSFM accuracy[2,3]. Further reducing the step-size and increasing the simulated bandwidth by increasing the numbers of samples per symbol did not impact the displayed results, which indicates 0.1 km step-size and 16 samples per symbol are accurate enough for the systems in this paper. For the RP on $\gamma$ model, the integral of Eq. (17) was evaluated numerically using an integration step of also 0.1 km. For the RP on $\beta_2$, the derivatives were obtained in the frequency domain. The dispersion operator was ideally implemented in the SSFM and in all simulated equations in the frequency domain. The CDC was implemented in the frequency domain with the dispersion operator applied with negative fibre length. Before the matched filter, the signal was scaled with the inverse scaling factor used in the transmission to adjust the input power. The matched filter was the same RRC filter as used in the transmission.

## Data availability

The data that support the findings of this study are available from the corresponding author (E.A.) upon request.

## Code availability

The computer codes used to generate the results in this manuscript are available from the corresponding author (E.A.) upon request.

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

## Acknowledgements
We thank Dr. Tobias Fehenberger for fruitful discussions about the RP on *γ*. We also thank Dr. Nicola Calabretta and Dr. Domaniç Lavery for insightful discussions about passive optical networks, and Dr. Gabriele Liga for interesting discussions about optical fibre models. This work is supported by the Netherlands Organisation for Scientific Research (NWO) via the VIDI Grant ICONIC (project number 15685). The work of A. Alvarado has received funding from the European Research Council (ERC) under the European Union's Horizon 2020 research and innovation programme (grant agreement No. 757791). The work of E. Agrell has received funding from the Swedish Research Council (VR) under Grant no. 2017-03702.

## Author contributions
E.A. proposed the idea of applying RP on the linear coefficient. V.O. implemented the codes that generated the figures and obtained the final equations for the new model. E.A. solved Supplementary Equation (41) for the lossless case, which was fundamental for the model derivation. A.A. provided crucial guidance and support for preparing the paper, both on its content and structure. V.O. wrote the paper and all the other authors contributed with substantial reviews and comments.

## Competing interests
The authors declare no competing interests.
