## [Peer Review File · Nature Communications]

Reviewers' comments:

Reviewer #1 (Remarks to the Author):

While the subject matter of this work, i.e., applying RP on dispersion coefficient instead of the nonlinear coefficient is counter-intuitive and theoretically appealing, I am not convinced about the true added value of the proposed alternative approach. The main reason is that, the authors quantify the accuracy only in terms of NSD and consider only single-span scenario, and do not include receiver DSP and SNR. This might be enough for people working on applied mathematics, and numerical solutions to PDE, but engineers are demanding! I would like to see SNR vs. launched power curves, produced by RPs on dispersion, and Kerr nonlinear effect, as well as the optimized split-step simulations, for multi-span full transmission, for SSMF and NZDSF fibers at two baud-rates, in order to appreciate the operational value of the new approximation. It is up on authors to include matched filtering and compute the sampled signal nonlinear variance using their new approximation for the field. Otherwise, the current paper, is not suitable to be published in Nature. I highlight the fact that the paper is very well-written, and has the potential to be accepted, but without a concrete operational demonstration of the usefulness of the new proposed method in modeling multi-span fiber-optic systems, and without comparing the error-bars on SNR (and not NSD) the paper is not solid enough for Nature.

One formal comment: please remove all 'Supplementary Equation' phrases in the supplementary notes. Equation numbers are enough. I found it very annoying

Reviewer #2 (Remarks to the Author):

This article addresses a fundamental problem with considerable practical implications: modelling of propagation of communications signals in the optical fiber infrastructure. The infrastructure that is indeed supporting all the Internet data exchange, besides the few last hundreds of meters covered by wireless data transport. So, this is not only a theoretical problem, as it has several practical implications of considerable interest for the scientific community, as well as for the commercial operators and vendors operating in the field.

The authors present properly the problem, and the original results are well supported and very well presented. So, from a formal point of view, my judgement on the article is very positive. But, as previously stated, this is not a theoretical problem far from practical applications, so the authors cannot avoid to including in their interesting analysis some additional degrees of realism. Because of this, I invite the authors to revise the manuscript by addressing the specific concerns listed in the following.

1. The authors refer the entire analysis to the solution of the NLSE, asserting that it is the propagation equation for the modal amplitude in the single-mode fiber. It is only partially true. In case of targeting propagation of telecommunications signals, the propagation equation in the optical fiber is indeed the coupled NLSE – a stochastic equation--, that includes the random birefringence characterizing all the optical fibers used for telecommunications. And consequently, governs the evolution of the Jones vector, dual-polarization, modal amplitude. Moreover, depending on the nature of the considered propagating signal(s), the problem can be averaged with respect to the polarization rotations, obtaining the Manakov equation; or the Manakov-PMD equation, in case the PMD effect is included. Therefore, I'd suggest the authors to address and comment on this, by extending their analysis or at least by addressing whether their simplified approach limits the applicability of the proposed results to more realistic scenarios.

2. The authors derive the proposed theoretical analysis and compare the results to SSFM simulations. To this regard I have two concerns.

- a. The authors aim at proposing an innovative closed form approximated solution for the modal amplitude propagation in the NLSE for telecommunication signals. So, the validation cannot be limited to a single-channel single-modulation-format comparison. Exploitation of optical fibers for

telecommunications purposes relies on the wavelength division multiplexing, i.e., on the transmission of several dual-polarization and statistical independent signals on different center frequency. Filling the entire C-band and beyond. Therefore, the authors should address whether the proposed results can be applied to a modal amplitude that is indeed the sum of several spectrally separated signals: a WDM comb. Also, by addressing limitations given by the main wide-band effect taking place in the fiber: the power transfer enabled by the stimulated Raman scattering. Different modulation format must be considered as well.

b. The SSFM validations have been done using a simple matlab code based on a constant step p of 0.1 km. According to the authors' statement. It is well known, that the SSFM is a powerful yet delicate numerical tool, that may generate realistic yet unreal effects, depending on the joint choice of the step-size and simulation bandwidth, given the signal nature, the power and the fiber characteristics. So, the authors must better motivate the accuracy and the parameter choice for the SSFM tool, by properly referring to the extensive literature addressing the problem.

3. The authors address the problem of lossy fiber and lossless fiber. The latter is completely unrealistic, and it is not adequate for an article targeting the modeling of a realistic scenario. It should be much more adequate to address the problem of fibers including distributed amplification enabled by the stimulated Raman scattering.

Reviewer #3 (Remarks to the Author):

The paper develops a first-order perturbation model for optical fiber, where the perturbation is on the dispersion parameter. Theorem 2, which is the main result, seems to be novel and useful; past research has focussed on a perturbation on the non-linearity, see Theorem 1. The method used for analysis is the same as in past work, see the supplementary notes.

The rest of the paper compares the accuracy of the different models mainly for NZDSF fiber that has low dispersion (there are also comparisons to SSMF in the O-band in the ECOC paper). Not surprisingly, when the dispersion is low perturbing the dispersion is better than perturbing the non-linearity. However, it is interesting to see how the accuracy behaves with the launch power.

The results should be of interest to other researchers in the community, and they should motivate more work on improving the accuracy of "simple" models, and on further simplifying fiber models.

I list a few suggestions for improvement.

- One limitation of these models is that they neglect noise, which is not realistic. This should be emphasized up front in the abstract, and not only in the discussion section. Some discussion on how to include noise would be welcome, as well as references to papers that have done this (I think there are a few).

- The model requires more work to be useful, as the complexity of the computations is similar to that of the finite-difference method to solve the NLSE. For example, in the ECOC paper, the integral in z in (7) with the trapezoidal rule requires computing $F(t,z)$ at many points ("steps") in the fiber, and in each step computing $F(t,z)$ requires a nonlinear and a linear step very similar to those of the finite-difference method. This motivates adding discussion on computational complexity. If the complexity is similar, then also discussion on why one should use an approximate model, rather than the exact model, would be welcome.

- To make the model useful for estimating capacity, one might need to find a way to separate the contributions of individual pulses to turn the model into a discrete channel.

Reply to reviewers' comments

We thank all the reviewers for carefully reading the manuscript and for their comments, which resulted in significant and important improvements to the paper. We did our best to address all their comments and suggestions, while in the same time trying to stay within the maximum number of figures and words imposed by Nature. All changes in the manuscript are highlighted in red.

1 Main changes in the manuscript

- Three new sections were added in Results:
 - ‘Fibre length versus symbol rate’;
 - ‘Modulation format impact’;
 - ‘Discrete-time performance’;
- Methods section was adapted to include new details about the new simulations to address the reviewers comments;
- Fig. 5 was removed to save space and to avoid non-practical (theoretical) discussions;
- Fig. 7 and Fig. 8 were merged into ‘Fig. 6’;
- New figures:
 - ‘Fig. 7: NSD for RP on β_2 for different values of fibre length and distance. The system consists of a 80 km NZDSF with fixed input power 5 dBm.’;
 - ‘Fig. 8: Impact of QPSK, star 8-QAM, and 64-QAM modulation formats on the NSD versus input power. The system consists of an 80 km NZDSF with 10 Gbaud symbol rate.’;
 - ‘Fig. 9: SNR versus input power for two different symbol rates (10 and 40 Gbaud), and received constellations for SSFM, RP on γ , and RP on β_2 , with their respective PDFs. On the SNR results, receivers

with (w/) and without (w/o) CDC were considered for a QPSK modulation over a 20 km NZDSF. The constellations were obtained for 40 Gbaud with CDC at 16 dBm. **a** SNR results for the three models at a symbol rate of 10 Gbaud. **b** SNR results for the three models at 40 Gbaud. **c** Received RP on β_2 constellation and respective PDF. **d** Received SSFM constellation and respective PDF. **e** Received RP on γ constellation and respective PDF.’;

- ‘Fig. 2’: was adapted to include the discrete-time analysis.

As a general comment, we appreciate the reviewers’ numerous suggestions for extending the model, which would make the proposed model more general and more practically useful. As pointed out, such extensions may include noise, Raman amplification, dual-polarization, discrete-time, multiple spans, and WDM. We believe, however, that it is infeasible to rigorously cover all such extensions in a single paper within the word and figure limitations of Nature Communications. We hope that the editor and reviewers agree.

An analogous situation arose when the RP on γ model was first proposed. In their pioneering 2002 paper [20], Vannucci et al. studied a basic single-polarization single-span continuous-time NLSE equation, applied in a noiseless multi-span link. The above-mentioned extensions were added in future papers, thus gradually making the RP on γ model more general and more practically useful. In retrospect, it would have been impossible to condense all those contributions into a single paper.

Reply to Reviewer 1

While the subject matter of this work, i.e., applying RP on dispersion coefficient instead of the nonlinear coefficient is counter-intuitive and theoretically appealing, I am not convinced about the true added value of the proposed alternative approach. The main reason is that, the authors quantify the accuracy only in terms of NSD and consider only single-span scenario, and do not include receiver DSP and SNR. This might be enough for people working on applied mathematics, and numerical solutions to PDE, but engineers are demanding! I would like to see SNR vs. launched power curves, produced by RPs on dispersion, and Kerr nonlinear effect, as well as the optimized split-step simulations, for multi-span full transmission, for SSMF and NZDSF fibers at two baud-rates, in order to appreciate the operational value of the new approximation.

We thank the reviewer for the suggestions. We also agree that a further analysis in terms of SNR should be included. For that purpose, we added a whole new section in the paper called ‘Discrete-time performance’ (Sec. 2.11). In this section, we include:

- SNR results for SSFM, RP on γ , and RP on β_2 in a NZDSF;

- Two different symbol rates (10 and 40 Gbaud);
- Two types of receivers: with and without chromatic dispersion compensation;
- Analysis of the received constellation.

It is up on authors to include matched filtering and compute the sampled signal nonlinear variance using their new approximation for the field.

We thank the reviewer for the suggestion. Since there is no ASE noise, the SNR results with CDC are accounting for the nonlinear signal-signal noise, and is closely related to the nonlinear variance. Therefore, we chose to contain the discrete results to SNR only. On the other hand, matched filter and sampling were included in the ‘Discrete-time performance’ section (Sec. 2.11).

Otherwise, the current paper, is not suitable to be published in Nature. I highlight the fact that the paper is very well-written, and has the potential to be accepted, but without a concrete operational demonstration of the usefulness of the new proposed method in modeling multi-span fiber-optic systems, and without comparing the error-bars on SNR (and not NSD) the paper is not solid enough for Nature.

We appreciate the reviewer’s viewpoint on the paper. As explained above, we now include an extensive evaluation of SNR, which is a concrete demonstration of the usefulness of our model.

For the comment on multi-span full transmission, we emphasized in the discussion section of the manuscript that this is a limitation of the model (since it would result in high accumulated dispersion). In addition, we would like to point out that there are other optical communication systems that would benefit from the proposed model, such as PONs. To clarify this, we added the following explanation in the Discussion (Sec. 3):

As all simplified models, the proposed model has a limited range of validity. At the moment, the main applicability of the model is for applications that rely on low bandwidths (below 11 GHz) and short distances (below 80 km). This includes, for example, passive optical networks [27]. Another short-distance low-bandwidth application is hybrid fibre coax systems [28]. The model is in its present form not intended for long-haul or wavelength-division-multiplexed systems.

Two new references were added:

[27] Xue, L., Yi, L., Li, P. & Hu, W. 50-Gb/s TDM-PON based on 10G-class devices by optics-simplified DSP. In Proc. Opt. Fib. Commun. Conf. and

Exhib. (OFC) (2018).

[28] Wang, J. et al. Delta-sigma digitization and optical coherent transmission of DOCSIS 3.1 signals in hybrid fiber coax networks. *J. Lightw. Technol.* **36**, 568–579 (2018).

Furthermore, the fibre length and bandwidth limitation are also now explained in the new section ‘Fibre length versus symbol rate’ (Sec. 2.9).

One formal comment: please remove all ‘Supplementary Equation’ phrases in the supplementary notes. Equation numbers are enough. I found it very annoying

We agree with the reviewer that the repetition of the term ‘Supplementary Equation’ can be unpleasant. Unfortunately, this is part of the submission guidelines from Nature Communications.

Reply to Reviewer 2

This article addresses a fundamental problem with considerable practical implications: modelling of propagation of communications signals in the optical fiber infrastructure. The infrastructure that is indeed supporting all the Internet data exchange, besides the few last hundreds of meters covered by wireless data transport. So, this is not only a theoretical problem, as it has several practical implications of considerable interest for the scientific community, as well as for the commercial operators and vendors operating in the field. The authors present properly the problem, and the original results are well supported and very well presented. So, from a formal point of view, my judgement on the article is very positive.

We thank the reviewer for these kind words.

But, as previously stated, this is not a theoretical problem far from practical applications, so the authors cannot avoid to including in their interesting analysis some additional degrees of realism. Because of this, I invite the authors to revise the manuscript by addressing the specific concerns listed in the following.

1. *The authors refer the entire analysis to the solution of the NLSE, asserting that it is the propagation equation for the modal amplitude in the single-mode fiber. It is only partially true. In case of targeting propagation of telecommunications signals, the propagation equation in the optical fiber is indeed the coupled NLSE – a stochastic equation –, that includes the random birefringence characterizing all the optical fibers used for telecommunications. And consequently, governs the evolution of the Jones vector, dual-polarization, modal amplitude. Moreover, depending on the nature of the considered propagating signal(s), the*

problem can be averaged with respect to the polarization rotations, obtaining the Manakov equation; or the Manakov-PMD equation, in case the PMD effect is included. Therefore, I'd suggest the authors to address and comment on this, by extending their analysis or at least by addressing whether their simplified approach limits the applicability of the proposed results to more realistic scenarios.

We agree with the reviewer that the used NLSE can impose limits to the fibre effects covered by the model. We clarify this fact in a new paragraph after the description of the NLSE (Sec. 2.1):

This version of the NLSE disregards dual-polarization effects, such as polarization-mode dispersion, as well as other effects like stimulated Raman scattering. The latter can be reasonably ignored given the low bandwidth scenarios studied in this paper [1]. We chose to analyse the single-polarization equation as a first step towards RP on β_2 . Nevertheless, single-polarization transmission is still attractive for low cost optical systems [24].

One more reference was added:

[24] Chen, X. et al. 218-Gb/s single-wavelength, single-polarization, single-photodiode transmission over 125-km of standard single mode fiber using Kramers-Kronig detection. In Proc. Opt. Fib. Commun. Conf. and Exhib. (OFC) (2018).

We also state in the Discussion section that we consider the dual-polarization extension as a future work:

This paper represents the first steps in the theory of a new model. Possible extensions of this work are designing a receiver based on the regular perturbation on β_2 , deriving higher-order perturbations of the model, and adding noise within the regular perturbation analysis. The derivations were conducted for a single-polarization system, and the equations for dual-polarization are still a subject of further investigation.

2. The authors derive the proposed theoretical analysis and compare the results to SSFM simulations. To this regard I have two concerns.

a. The authors aim at proposing an innovative closed form approximated solution for the modal amplitude propagation in the NLSE for telecommunication signals. So, the validation cannot be limited to a single-channel single-modulation-format comparison. Exploitation of optical fibers for telecommunications purposes relies on the wavelength division multiplexing, i.e., on the transmission of several dual-polarization and statistical independent signals on different center frequency. Filling the entire C-band and beyond. Therefore, the authors should address whether the proposed results can be applied to a modal

amplitude that is indeed the sum of several spectrally separated signals: a WDM comb. Also, by addressing limitations given by the main wide-band effect taking place in the fiber: the power transfer enabled by the stimulated Raman scattering.

We agree with the reviewer that many communication systems rely on WDM transmission and many different modulation formats. To address this issue, we explain in the discussion section that this is a current drawback of this model and that there are other systems that does not rely on large bandwidths and long distances. Please see the response for Reviewer #1 in page 3.

Filling the entire C-band, using WDM with large bandwidths per channel or considering long-distance transmission would result in high accumulated dispersion. This is a regime where our model is not accurate anymore (dispersion cannot be treated as a perturbation). To further clarify this, we added a new section ‘Fibre length versus symbol rate’ (Sec. 2.9).

Different modulation format must be considered as well.

Done. We added a new section called ‘Modulation format impact’ (Sec. 2.10). In addition, the new section ‘Discrete-time performance’ considers QPSK modulation.

b. The SSFM validations have been done using a simple matlab code based on a constant step p of 0.1 km. According to the authors’ statement. It is well known, that the SSFM is a powerful yet delicate numerical tool, that may generate realistic yet unreal effects, depending on the joint choice of the step-size and simulation bandwidth, given the signal nature, the power and the fiber characteristics. So, the authors must better motivate the accuracy and the parameter choice for the SSFM tool, by properly referring to the extensive literature addressing the problem.

We agree with the reviewer that the SSFM is very delicate depending on the signal bandwidth, power, and type of fibre. We have double-checked all our simulations by reducing the step-size both in time and spatial domain and observed no differences. To clarify this, we now state in Section 4.1 (Methods) that the chosen step-sizes (time and space) are accurate enough for our simulations:

The step-size and the simulation bandwidth substantially impact the SSFM accuracy [2,3]. Further reducing the step-size and increasing the simulated bandwidth by increasing the numbers of samples per symbol did not impact the displayed results, which indicates 0.1 km step-size and 16 samples per symbol are accurate enough for the systems in this paper.

One more reference was added:

[3] Bosco, G. et al. Suppression of spurious tones induced by the split-step method in fiber systems simulation. *J. Lightw. Technol.* 12, 489–491 (2000).

3. *The authors address the problem of lossy fiber and lossless fiber. The latter is completely unrealistic, and it is not adequate for an article targeting the modeling of a realistic scenario. It should be much more adequate to address the problem of fibers including distributed amplification enabled by the stimulated Raman scattering.*

We thank the reviewer for the observation. We have changed how we present the results from “lossless fibres” to “ideal distributed amplification”. Now, we compare *amplification schemes* instead of “fibre types”, which is a more realistic approach. The following changes were made in Sec. 2.5:

In systems with distributed Raman amplification, the power profile is approximately flat [26] and the attenuation term in (1) is often neglected. In this case, a simpler analytical form for $B(t, z)$ when compared to equation (22) is achieved. This simplification is given in the next theorem.

Theorem 3. *With ideal distributed amplification, the functions A_0 and A_1 in equation...*

One new reference was added:

[26] Vasilyev, L. Raman-assisted transmission: toward ideal distributed amplification. In *Proc. Opt. Fib. Commun. Conf. and Exhib. (OFC)* (2003).

We also removed Fig. 5, which described the power gain between “lossy” and “lossless” fibres. We believe the discussion was rather theoretical than related to a practical problem.

Reply to Reviewer 3

The paper develops a first-order perturbation model for optical fiber, where the perturbation is on the dispersion parameter. Theorem 2, which is the main result, seems to be novel and useful; past research has focussed on a perturbation on the non-linearity, see Theorem 1. The method used for analysis is the same as in past work, see the supplementary notes.

The rest of the paper compares the accuracy of the different models mainly for NZDSF fiber that has low dispersion (there are also comparisons to SSF in the O-band in the ECOC paper). Not surprisingly, when the dispersion is low perturbing the dispersion is better than perturbing the non-linearity. However, it is interesting to see how the accuracy behaves with the launch power.

The results should be of interest to other researchers in the community, and they should motivate more work on improving the accuracy of “simple” models,

and on further simplifying fiber models.

We thank the reviewer for these kind words.

I list a few suggestions for improvement.

- One limitation of these models is that they neglect noise, which is not realistic. This should be emphasized up front in the abstract, and not only in the discussion section. Some discussion on how to include noise would be welcome, as well as references to papers that have done this (I think there are a few).

Thank you for the suggestion. We now emphasized this in the abstract:

Here, we present a new approximate model for the nonlinear optical fibre channel in the weak-dispersion regime, in a noiseless scenario.

In the Discussion section (Sec. 3), we also mentioned that adding noise to the model is left as future work and added some discussion on how noise was added in RP on γ :

Another drawback of the proposed model is that it neglects the noise. This effect has been considered in the literature for the regular perturbation on γ in [22], where noise was added in the zeroth order linear equation, followed by a Karhunen–Loève expansion to account for nonlinear signal–noise interactions. The same approach for the regular perturbation on β_2 would lead to cumbersome equations, since the zeroth order equation is nonlinear in this case.

This paper represents the first steps in the theory of a new model. Possible extensions of this work are designing a receiver based on the regular perturbation on β_2 , deriving higher-order perturbations of the model, and adding noise within the regular perturbation analysis. . . .

- The model requires more work to be useful, as the complexity of the computations is similar to that of the finite-difference method to solve the NLSE. For example, in the ECOC paper, the integral in z in (7) with the trapezoidal rule requires computing $F(t,z)$ at many points ("steps") in the fiber, and in each step computing $F(t,z)$ requires a nonlinear and a linear step very similar to those of the finite-difference method. This motivates adding discussion on computational complexity. If the complexity is similar, then also discussion on why one should use an approximate model, rather than the exact model, would be welcome.

Here we humbly have to disagree. The integral in (7) of the ECOC submission, which is (43) in the Supplementary Notes, was calculated analytically using Theorem 2 in this manuscript. This is done only once for the whole length of the fibre, with no need to split it in many spatial steps. Indeed, this is one of the main advantages with the proposed model. The following statement was

added in Section 2.5 to address the reviewer's comment:

The functions A_0 and A_1 depend on elementary functions of z (see equations (13), (27), (28), and (29)). This results in the same calculation time for A_M at any distance z , in contrast to the SSFM, for which the number of necessary steps increases with the distance. For this reason, RP on β_2 calculations are significantly faster than SSFM calculations for the parameters simulated in this paper.

- To make the model useful for estimating capacity, one might need to find a way to separate the contributions of individual pulses to turn the model into a discrete channel.

Partly done. We included a discrete numerical analysis by applying matched filtering and sampling and extracting the received symbols corresponding to individual transmitted pulses. Based on the received symbols, we calculate the SNR to analyse the model performance. This discrete-time processing is now discussed in Section 2.1, where Figure 2 has been significantly revised. SNR results are now reported in the new Section 2.11 and Figure 10.

The derivation of an analytical, symbol-timed RP on β_2 model is left as future work, as added in Section 3:

The derivations were conducted for a single-polarization system, and the equations for dual-polarization are still a subject of further investigation. Separating the contribution of individual pulses and finding a discrete model are also left as future work. . . .

REVIEWERS' COMMENTS:

Reviewer #1 (Remarks to the Author):

No further comments. It can be accepted.

Reviewer #3 (Remarks to the Author):

I have no further comments, the authors have addressed my concerns.

Reply to reviewers' comments

We thank all the reviewers for carefully reading the manuscript and for their comments, which resulted in significant and important improvements to the paper. Since there were no further comments for the second round of reviews, we did not add more replies.

Reviewer 1 (remarks to the author)

No further comments. It can be accepted.

Reviewer 3 (remarks to the author)

I have no further comments, the authors have addressed my concerns.